# Anti-Inflammatory Effect on Colitis and Modulation of Microbiota by Fermented Plant Extract Supplementation

**Manabu Sugimoto** [1],*, **Toshiro Watanabe** [2], **Motoko Takaoka** [3], **Kyoko Suzuki** [3], **Tadatoshi Murakami** [4], **Nobutada Murakami** [4] and **Shoichi Sumikawa** [4]

1   Institute of Plant Science and Resources, Okayama University, 2-20-1 Chuo, Kurashiki, Okayama 710-0046, Japan
2   Department of Food and Nutrition, Sonoda Women's University, 7-29-1 Minamitsukaguchi, Amagasaki, Hyogo 661-8520, Japan; watnb-ts@sonoda-u.ac.jp
3   Department of Biosphere Sciences, Kobe College, 4-1 Okadayama, Nishinomiya, Hyogo 662-8505, Japan; takaoka@mail.kobe-c.ac.jp (M.T.); suzuki@mail.kobe-c.ac.jp (K.S.)
4   Functional Food Creation Research Institute Co., Ltd., 4422-1 Yoshikawa, Kibichuo, Kaga, Okayama 716-1241, Japan; murakami@kinousei.co.jp (T.M.); n-murakami@kinousei.co.jp (N.M.); sumikawa@kinousei.co.jp (S.S.)
*   Correspondence: manabus@okayama-u.ac.jp; Tel.: +81-86-424-1661

**Abstract:** Although results of recent studies suggest that fermented foods strongly affect the gut microbiota composition and that they relieve inflammatory bowel disease symptoms, some reports have described that fermented foods increase some inflammation markers based on differences in fermented food materials. This study evaluated the effects of fermented plant extract (FPE) on dextran sulfate sodium (DSS)-induced colitis in mice and the effects on fecal microbiota composition in humans. Mice fed 5% FPE with 3% DSS (FPE group) showed no body weight loss, atrophy of colonic length, or bloody stool, similar to mice fed a basal diet (negative group), whereas mice fed 3% DSS (positive group) exhibited those effects. Concentrations of inflammation markers IL-6 and TNF-$\alpha$ were not significantly different between FPE and negative groups; however, those concentrations became higher in the positive group. 16S ribosomal RNA gene sequencing was used to characterize fecal microbiota in healthy women before and after 3-month FPE supplementation. The FPE supplementation induced increases in *Firmicutes* phyla and in *Clostridiales* order, which play a central role in inflammation suppression. These results suggest that FPE enhances *Clostridiales* growth in the gut and that it has an anti-inflammatory effect.

**Keywords:** fermented plant extract; microbiota; dextran sulfate sodium; inflammatory; *Clostridiales*



## 1. Introduction

Inflammatory bowel disease, as typified by diseases such as ulcerative colitis and Crohn's disease, affects the gastrointestinal tract. No satisfactory treatment has been found because its causes remain unidentified. Moreover, the Japanese Ministry of Health, Labor, and Welfare has specified these diseases as intractable diseases. The number of patients with these diseases has been increasing in Japan [1,2]. Lack of dietary fiber and the consumption of western foods are implicated as reasons. Actually, dietary changes alter gut microbiota composition. In fact, dysbiosis and disruption of immunological homeostasis have been associated with inflammatory diseases [3–13] because the gut microbiota affects immunologic, nutritional, and metabolic processes of the human body [14], indicating that diet strongly affects inflammatory bowel disease.

Functional foods are part of the human diet. They have been demonstrated to provide health benefits and to decrease the risk of chronic diseases beyond those provided by adequate nutrition [15]. Functional foods include naturally occurring bioactive substances, supplemented bioactive substances, and derived food ingredients. Especially, specialized

ingredients in functional foods promoting the growth or activity of specific bacteria, such as *Lactobacillus* and *Bifidobacterium* genera, have been specifically examined as prebiotics. Recent scientific advances show that modifying the bacterial composition of the intestinal ecosystem induces functional changes such as the host physiology [16–19]. A diet with greater amounts of dietary fiber increases the composition of *Clostridiales* in mice. This is a dominant class of commensal microbes that produces butyric acid to induce colonic regulatory T (Treg) cells, which play a central role in suppressing inflammatory and allergic responses [20–22]. Therefore, it is crucially important to evaluate foods to ascertain whether they increase the composition of bacteria that produce bioactive molecules.

Fermented foods have been a fundamentally important part of the human diet for centuries in most parts of the world. Fermented foods of many types are produced from animal and plant materials, some of which have the potential to provide additional health benefits through fermentation as functional foods [23,24]. Some fermented foods have beneficial immune, glycemic, and anti-inflammatory activities [25–28], whereas other fermented foods increase some inflammation markers [29]. These results indicate the necessity for assessment of each fermented food to assess its health-promoting activities because of differences in the materials and microbes used for fermentation.

For this study, we used a fermented plant extract (FPE) produced from whole plants to examine the effects on dextran sulfate sodium (DSS)-induced colitis in a mouse model. Bloody stools and atrophy of colonic length were not observed. The TNF-α and IL-6 concentrations were suppressed by FPE supplementation. Subsequently, 16S rRNA amplicon sequencing revealed that the composition of *Clostridiales* order had increased significantly in the fecal samples of young women after 3-month FPE supplementation with daily diet.

## 2. Materials and Methods

### 2.1. Preparation of Fermented Plant Extract

Whole fruits and vegetables with peels and seeds were minced and extracted using muscovado sugar in separate barrels (Table S1). For animal experimentation, the extracts were mixed with mushroom, pulse, cereals, seaweed, and decoction of loquat leaves, to make a starting material of 75 kinds, consisting of 50.0% of sugar, 23.9% of fruits, 15.6% of vegetables and wild herbs, 2.0% of mushrooms, 1.8% of seaweed, and 6.7% of pulse and cereals. Lactic acid bacteria, *Lactiplantibacillus pentosus*, *L. plantaru*, *Pediococcus pentosaceus*, *P. acidilactici*, *Lacticaseibacillus paracasei*, *L. casei*, *Lactococcus lactis*, *Latilactobacillus curvatus*, *Leuconostoc mesenteroides*, and *Levilactobacillus brevis* were added to the starting material and fermented at room temperature. After three year-fermentation and maturation, the fermented extract was strained and heated at 80 °C for 10 min to obtain FPE. pH, water content, and brix of were 3.8–4.8, 33–39%, and 59–65%, respectively. For microbiota analyses, the materials, which were acceptable for exporting to international countries, were mixed to make a starting material of 40 kinds, consisting of 42.0% of sugar, 24.6% of fruits, 19.4% of vegetables and wild herbs, 5.0% of mushrooms, 3.0% of seaweed, and 6.0% of pulse and cereals. The mixture was fermented using the same lactic acid bacteria and method as that described above. pH, water content, and brix of the fermented extract were 3.8–4.5, 34–42%, and 57–62%, respectively. As an FPE package, 5 g of the mixture, consisting of 40% of FPE, 59% of apple extract, and 1% of plum juice, was packed in an aluminum bag. The mixture was prepared to consist of 43.8% of apple juice, 26.3% of sugar, 29.0% of dextrin, and 0.9% of caramel color. Then, 5 g of the mixture was packed in an aluminum bag as a placebo package.

### 2.2. Animals

Male BALB/c mice were purchased from CLEA Japan, Inc. (Tokyo, Japan). They were kept under standard conditions in a room at 21–24 °C with a constant 12 h light/dark cycle. The ethics committee of Sonoda Women's University approved the experiments, which were performed in accordance with relevant guidelines and regulations.

### 2.3. Induction of Colitis

Five-week-old mice were divided into three groups: a negative group ($n = 8$), a positive group ($n = 8$), and an FPE group ($n = 8$). The positive group was fed diet with 3% dextran sulfated sodium 5000 (DSS) solution. The FPE group was fed diet containing 5% FPE with 3% DSS solution. The negative group was fed diet with water for 7 days. The diet composition provided to each group is presented in Table S2.

### 2.4. IL-6 and TNF-α Assays

IL-6 and TNF-α concentrations in mouse sera were found using Mouse IL-6 Assay Kit (Immuno-Biological Laboratories Co., Ltd., Gunma, Japan) and Quantikine ELISA Mouse TNF-α Kit (R&D Systems, Minneapolis, MN, USA), respectively, according to the instructions of the manufacturers.

### 2.5. Dietary Supplementation and Fecal Collection

We recruited 20 healthy women (age of 20–24 years, $21.4 \pm 1.2$ years) from the School of Human Sciences, Kobe College. All participants gave written, informed consent before participating in the study. This study was approved by the review board of Kobe College. Participants were assigned randomly into two groups. Participants in the FPE group were administered one FPE package three times after meals every day for 3 months in addition to their habitual diet. Participants in the placebo group were administered one placebo package in the same way as that of FPE group. No dietary restriction was instructed on participants. Participants collected fecal materials before and after 3-month administration. The materials were suspended in a solution (DNA/RNA Shield$^{TM}$; Zymo Research Corp., Irvine, CA, USA).

### 2.6. Microbial DNA Extraction

Total microbial DNA was extracted from each fecal suspension using a kit (Quick-DNA$^{TM}$ Fecal/Soil Microbe Miniprep; Zymo Research Corp., Irvine, CA, USA) according to the manufacturer's instruction. The DNA samples were examined using a spectrophotometer (Nanodrop ND-1000; Thermo Scientific, Waltham, MA, USA) by measuring absorbance values at 260 and 280 nm.

### 2.7. 16S rRNA Gene Sequencing

The V3–V4 region of the microbial 16S rRNA gene in each DNA sample was amplified by PCR using the universal primer set 341F/806R according to a protocol described elsewhere [30]. A dual-indexing amplification and sequencing approach was used. The resulting amplicons were purified and subjected to sequencing with $2 \times 301$ bp paired-end reads on the MiSeq systems (Illumina, Inc., San Diego, CA, USA) using MiSeq v3 reagent kit (Illumina) according the protocols described by Illumina (https://jp.support.illumina.com/content/dam/illumina-support/documents/documentation/chemistry_documentation/16s/16s-metagenomic-library-prep-guide-15044223-b.pdf, accessed on 5 April 2021). The raw sequence data quality was checked using the FastQC quality-control tool (Babraham Bioinformatics, Cambridge, United Kingdom; http://www.bioinformatics.babraham.ac.uk/projects/fastqc/, accessed on 5 April 2021). The datasets were analyzed with QIIME 1.9.1 pipeline (Quantitative Insights Into Microbial Ecology; http://qiime.org, accessed on 5 April 2021) using GreenGenes 13.5 [31].

### 2.8. Statistical Analysis

Data analysis was performed using the analysis of Bonferroni's multiple comparison test and Student's *t*-test, with a statistical significance set at $p < 0.05$.

## 3. Results and Discussion

### 3.1. Anti-Colitis Effect of FPE in Mice

Plant materials of fermented foods, such as asparagus, garlic, chicory, onion, wheat, barley, rye, soybean, peas, beans, banana, tomato, seaweeds, and microalgae, contain lactulose, galactooligosaccharides, fructooligosaccharides, inulin, maltoorogosaccharides, and resistant starch, known as ''prebiotics'', adversely affect the growth and activity of specific bacteria producing immunomodulatory products, particularly a short-chain fatty acid such as butyric acid. In addition, fermentation of these materials produces novel bioactive compounds working as prebiotics [25–28,32]. Therefore, we specifically examine the potential of plant materials and the produced FPE. FPE produced from extract of 75 kinds was rich in phytochemicals such as dietary fiber and polyphenol, and physiological function such as anti-oxidative, anti-inflammatory, and anti-allergy activities [33].

To assess the anti-colitis effects of FPE, acute colitis was induced by DSS feeding for 7 days to evaluate the effects of FPE on gut inflammation. The daily dietary intakes of negative group, positive group, and FPE group, which were, respectively, $2.1 \pm 0.2$, $1.8 \pm 0.2$, and $2.0 \pm 0.3$ g/day, were not found to be significantly different. The positive group was found to have significantly reduced body weight after 7-day DSS administration ($21.9 \pm 0.31$ g) compared with results found for the negative group ($24.6 \pm 0.48$ g), although the FPE group was found to have no significant difference in body weight ($23.5 \pm 0.41$ g) compared with the negative group (Figure S1). The bloody stool appeared at 5 days after DSS administration in the positive group. Its color was brown. It progressed at 6 and 7 days. The color was red. In contrast, the bloody stool found in the FPE group was not observed as it was in the negative group, whereas the fecal color of the FPE group was black (Table 1). The colon length in the positive group ($5.9 \pm 0.17$ cm) was significantly shorter than that in the negative group ($8.3 \pm 0.24$ cm), but the FPE group exhibited no difference in the colon length, which was $7.7 \pm 0.21$ cm compared with the negative group (Figure 1 and Figure S2). The concentrations of IL-6 and TNF-$\alpha$, inflammatory cytokines, and markers of inflammation were increased significantly, five-fold and four-fold, respectively, compared with the negative group after 5-day DSS administration. However, those in the FPE group were not significantly different from those in the negative group (Figure 2). These results demonstrate that our FPE has an anti-inflammatory effect on DSS-induced mouse colitis.

**Table 1.** Blood stool score of feces.

| Day after DSS Administration | Negative Group | Positive Group | FPE Group |
|:---:|:---:|:---:|:---:|
| 4 | - | - | - (BK) |
| 5 | - | + | - (BK) |
| 6 | - | ++ | - (BK) |
| 7 | - | ++ | - (BK) |

-, normal; +, brown bloody stool; ++, red bloody stool; BK, black color.

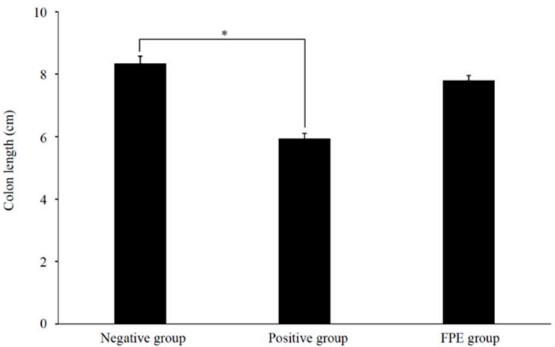

**Figure 1.** Colon length of mice after 7 day-administration. The values are expressed as means $\pm$ SEM ($n = 8$). * $p < 0.05$, accessed using Bonferroni's multiple comparison test.

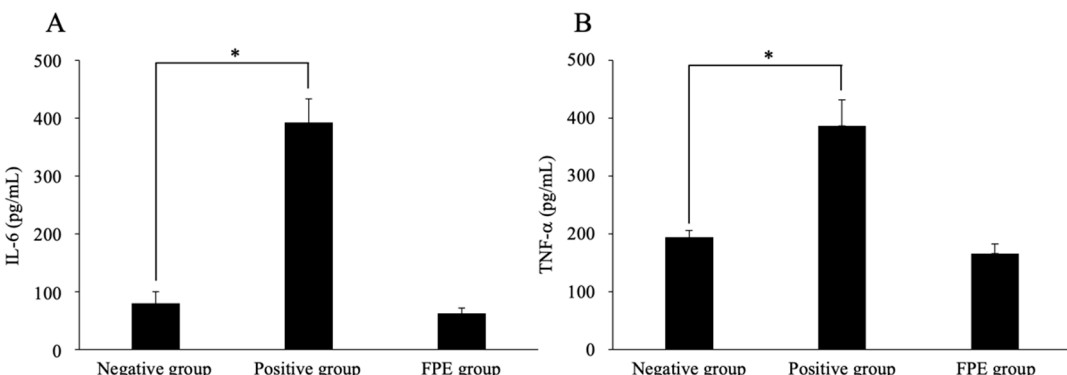

**Figure 2.** IL-6 (**A**) and TNF-α (**B**) concentrations of mice serum after 7-day administration. The values are expressed as means ± SEM (*n* = 8). * *p* < 0.05, assessed using Bonferroni's multiple comparison test.

*3.2. Change of Gut Microbiota Composition in Human*

To investigate changes of gut microbiota composition attributable to dietary supplementation of FPE, FPE was produced from extracts of 40 kinds, of which plant materials were included in those of 75 kinds and total content of fruits, vegetables, and wild herbs, which are sources of prebiotic [34], was 44% in extract of 40 kinds and dietary fiber was 4.6 g/100 g extract in FPE from extracts of 40 kinds, while total content of them in extract of 75 kinds was 39.5% and FPE from extract of 75 kinds contained 1.9 g dietary fiber/100 g extract. FPE from extract of 40 kinds contained same kinds of nutritional compounds such as amino acid and organic acid as FPE from extract of 75 kinds (Table S3). Each of 20 participants supplemented one FPE package or placebo package was administered three times after meals every day for 3 months. The changes of body weight, body fat mass, muscle mass, and body mass index (BMI) of the FPE group after 3-month administration were not significantly different from those of the placebo group (Figure S3). Based on data collected through a questionnaire, the scores of health conditions of both the FPE group and the placebo group after 3-month administration were not significantly different compared with those before administration. These results indicate that FPE supplementation did not affect health or physical condition.

Microbial DNA was isolated from fecal samples collected from FPE and placebo groups before and after 3-month administration. The DNA sequencing of the V3–V4 amplicons from 40 samples revealed 5,295,873 paired-end sequence reads with an average of 132,397 reads per sample. After filtering to eliminate poor quality sequences, finally 4,890,551 sequences with an average of 122,264 reads per sample were generated. The composition of bacterial phyla in feces of the FPE group before supplementation was not significantly different from that of the placebo group, of which *Firmicutes* was the most abundant, followed by *Bacteroidetes* and *Actinobacteria*. After 3 months of supplementation, the composition of *Firmicutes* in the FPE group increased along with reduction in *Bacteroidetes*. It was found to be significantly different from that in the placebo group (*p* = 0.001) (Table 2, Figure S4). Reportedly, dietary fiber can be designed to favor beneficial bacterial groups; certain *Firmicutes* access insoluble matrix fibers and resistant starch [35,36]. Plant materials in FPE, which contain fiber, starch, and oligosaccharides, can induce *Firmicutes* phyla in the gut. At the bacterial order level, the composition of *Clostridiales*, 39.0 ± 1.8%, increased significantly to 43.6 ± 2.5% after 3-month administration (*p* = 0.03) in the FPE group, whereas those in the placebo group before and after 3 months were 37.8 ± 1.6% and 38.0 ± 2.1%, respectively (Figure 3, Table S4). *Clostridiales* order in FPE group after 3-month administration consisted of 49% of *Lachnospiraceae* family and 33% of *Ruminococcaceae* family, which produce butyric acid [37,38]. *Clostridiales* induces colonic Treg cells and generates short-chain fatty acid, butyric acid, by fermentation of dietary fiber [39,40]. Butyric acid promotes to increase the expression of IL-10 in Treg cells and inhibits the activation of the transcription factor, NF-κβ, in gut cells. Increase of IL-10 in Treg cells

and decrease of NF-κβ lead the intestinal immune homeostasis, suppressing systemic and mucosal immune activation to control intestinal inflammation and contributing to maintaining tolerance towards gut microbiota [20–22,41]. Taken together, results show that FPE enhances *Clostridiales* growth in the gut and show that it has an anti-inflammatory effect on colitis.

**Table 2.** Microbiota composition at the phyla level in feces of placebo and fermented plant extract (FPE) groups before and after 3 month-administration (%).

| Phyla | Placebo Group | | FPE Group | |
|---|---|---|---|---|
| | 0 Month | 3 Months | 0 Month | 3 Months |
| Actinobacteria | $16.3 \pm 1.7$ | $15.8 \pm 1.7$ | $16.8 \pm 1.8$ | $15.5 \pm 1.1$ |
| Bacteroidetes | $36.1 \pm 1.3$ | $35.7 \pm 1.5$ | $32.2 \pm 2.6$ | $31.3 \pm 1.2$ |
| Cyanobacteria | 0 | 0 | $0.1 \pm 0.1$ | 0 |
| Firmicutes | $43.1 \pm 1.5$ | $42.6 \pm 2.1$ | $46.7 \pm 1.5$ | $50.2 \pm 1.5$ ** |
| Fusobacteria | $0.2 \pm 0.1$ | $0.1 \pm 0.1$ | $0.2 \pm 0.1$ | $0.1 \pm 0.1$ |
| Lentisphaerae | 0 | $0.1 \pm 0.1$ | $0.3 \pm 0.2$ | 0 |
| Nitrospirae | 0 | 0 | 0 | 0 |
| Proteobacteria | $3.7 \pm 1.6$ | $2.2 \pm 0.4$ | $3.7 \pm 0.9$ | $2.7 \pm 0.5$ |
| Spirochaetes | 0 | 0 | 0 | 0 |
| Synergistetes | 0 | 0 | 0 | 0 |
| TM7 | 0 | 0 | 0 | 0 |
| Tenericutes | 0 | $0.3 \pm 0.3$ | 0 | 0 |
| Verrucomicrobia | $1.5 \pm 1.4$ | $3.2 \pm 2.1$ | 0 | 0 |

** $p < 0.01$, assessed using Student's *t*-test.

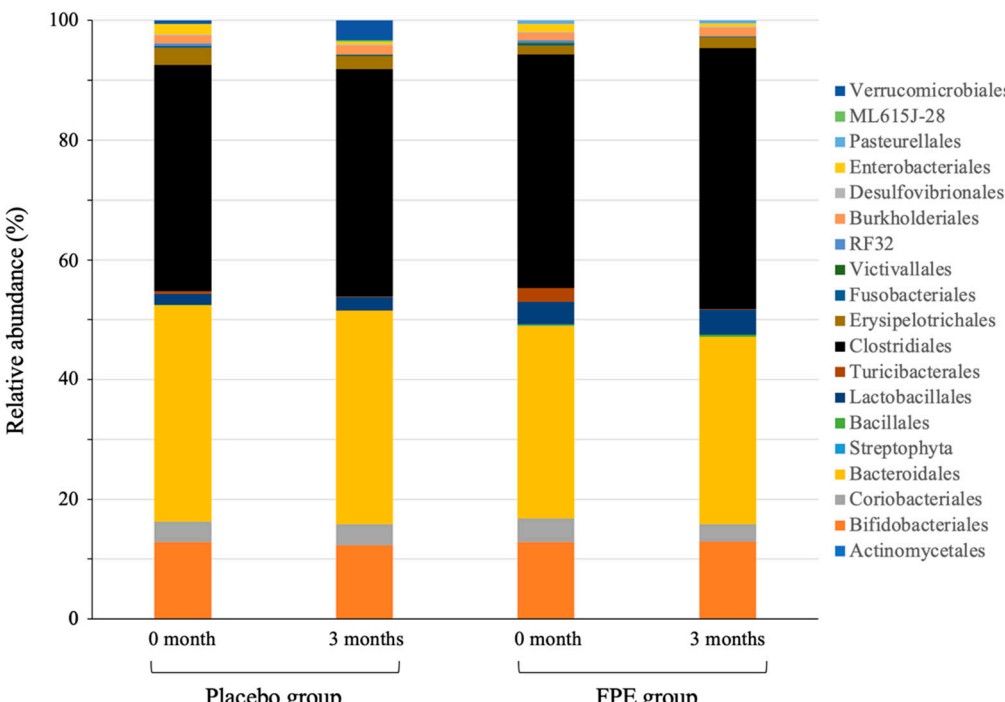

**Figure 3.** Microbiota composition at the order level in feces of placebo and FPE groups before and after 3-month administration.

Inflammatory bowel disease is generally treated with medicine. Lifestyle habits, particularly diet, have been implicated in the pathogenesis of inflammatory bowel disease [42]. Dietary therapy is assumed to lack life-threatening and severe side effects. However, a low-fat diet or a rich-fiber diet takes time and effort to prepare. Our study demonstrated that the daily diet supplemented with FPE directly prevented inflammation and changed the microbial composition in the gut, especially the increase of *Clostridiales* order, which

plays a central role in suppressing inflammation, suggesting that FPE could improve the quality of human life against inflammatory bowel disease.

## 4. Conclusions

The daily diet was supplemented with FPE. Mice fed FPE with DSS for 7 days showed no body weight loss, atrophy of colonic length, or bloody stool. The concentrations of inflammation markers IL-6 and TNF-α did not increase in the mice, demonstrating that FPE supplementation suppressed inflammation. The microbiota composition in feces from humans eating a diet supplemented with FPE for 3 months showed increased *Clostridiales* order, which produces short-chain fatty acids to suppress inflammation. These results indicate that FPE supplementation could enhance *Clostridiales* growth in the gut and indicate that it can have anti-inflammatory effects on colitis.

**Supplementary Materials:** The following are available online at https://www.mdpi.com/article/10.3390/fermentation7020055/s1. Table S1: Materials of FPE for animal and human. Table S2: Composition of experimental diet (%). Table S3: Nutritional compound and amino acid content in FPE produced from 40 kinds of extracts. Table S4: Microbiota composition at the order level in feces of placebo and FPE groups before and after 3 month-administration (%). Figure S1: Body weight of mice after 7-day administration. The values are expressed as means $\pm$ SEM ($n = 8$). * $p < 0.05$, assessed using Bonferroni's multiple comparison test. Figure S2: Colon length of mice after 7-day administration. Figure S3: Change of body weight (A), fat mass (B), muscle mass (C), BMI (D), and health condition (E) of FPE group or placebo group. Values are expressed as means $\pm$ SD ($n = 8$) assessed using Student's *t*-test. Figure S3: Microbiota composition at the phyla level in feces of placebo and FPE groups before and after 3-month administration.

**Author Contributions:** M.S.: Conceptualization, Formal analysis, Investigation, Methodology, Writing—original draft, review and editing. T.W.: Conceptualization, Formal analysis, Investigation, Methodology, Writing—review and editing. M.T.: Conceptualization, Formal analysis, Investigation, Methodology, Writing—review and editing. K.S.: Formal analysis, Investigation, Methodology. T.M.: Resources, Writing—review and editing. N.M.: Resources, Writing—review and editing. S.S.: Resources, Methodology. All authors have read and agreed to the published version of the manuscript.

**Funding:** This research received no external funding.

**Institutional Review Board Statement:** Ethical review and approval were waived for this study. It was conducted according to legally established internal rules based on research ethics recommendations and with the informed consent of all study participants.

**Informed Consent Statement:** Informed consent was obtained from all study participants.

**Data Availability Statement:** The data described as a result of this study are available in the article and Supplementary Materials.

**Conflicts of Interest:** The authors declare that they have no conflict of interest.

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
