# Peer review of "Anti-Inflammatory Effect on Colitis and Modulation of Microbiota by Fermented Plant Extract Supplementation"

_fermentation, doi:10.3390/fermentation7020055_

Round 1

Reviewer 1 Report

The research topic is very interesting and topical. Therefore, in my opinion, this manuscript should be published. Before being allowed to be published, the manuscript should be corrected and supplemented with missing information.

  • The introduction to the research topic is well prepared and contains up-to-date references. The aim of the research was clearly described.
  • Section "2.1. Preparation of Fermented Plant Extract" - please provide details of the fermentation process? what lactic acid bacteria were used? what were the conditions of inoculation? is it true that fermentation took three years? what were the fermentation parameters? what were the heating parameters of plant extract? what were the physic-chemical and microbiological characteristics of the obtained fermented plant extract? what was the acidity of the obtained fermented plant extract?
  • Lines 136-143 - are you sure your plant extracts contained these prebiotics?
  • In my opinion, the acidity, chemical composition and microbiological composition of the tested plant extract are important for the obtained results and their discussion. Therefore, when presenting the results, the real composition of the plant extract should be taken into account, based on the chemical analyzes or recipe calculations performed, and not data from references. The effect of the acidity of plant extract, its chemical and microbiological composition on the condition of the intestines and health parameters of the studied living organisms should be taken into account when discussing the results of the research.

Author Response

[Comment] Provide details of the fermentation process.

[Comment] Chemical composition and microbiological composition of the tested plant extract are important for the obtained results and their discussion.

[Answer] The authors thank the comment from the reviewer.The content of extract before fermentation, pH, water content, and brix of the extract after fermentation, and lactic acid bacteria used for fermentation were shown in text as according to your comment (Page 2, Line 74-89). Phytochemical function, physiological function, and nutritional compound of FPE produced from 75 kinds of extract reported previously were shown in text (Page4, Line 152-154). Those characteristics of FPE produced from 40 kinds of extract were not analyzed, however, 40 kinds of plant materials were included in FPE produced from 75 kinds of extract, same kinds of lactic acid bacteria were used for fermentation, and total content of fruits, vegetables, and wild herbs, which are sources of prebiotic, was 44 % in extract of 40 kinds and dietary fiber was 4.6 g /100 g extract in FPE from extracts of 40 kinds, which were similar to FPE produced from extract of 75 kinds. These results were shown in text (Page 5, Line 183-189). We showed nutritional compounds in FPE from extract of 40 kinds in Table S3. Our result that change of gut microbiota composition after 3 month-administration of FPE produced from extracts of 40 kinds suggests the physiological effect as same as FPE produced from extracts of 75 kinds and are useful to select the key plant materials for anti-inflammatory effect.

Reviewer 2 Report

The aim of this study was to evaluate the effects of fermented plant extract (FPE) on dextran sulfate sodium (DSS)-induced colitis in mouse and effects on fecal microbiota composition in humans. This study is potentially important because of a recent interest in the relationship between prebiotics intake and the anti-inflammatory effect in the colon. However, there were many critical deficits in this study. I have major concerns for the manuscript.

  1. The active components in the FPE such as soluble and insoluble dietary fibers and oligosaccharides have to be described in the table. Without the information of these components, we can’t do the follow-up examination.
  2. Because the materials of FPE for animal experiments and human intervention study are different, relationship between two experiments cannot be simply compared.
  3. Since the contribution of short chain fatty acids to the anti-inflammatory effect on colitis was discussed, the concentrations of fecal short chain fatty acids have to be shown in the table.
  4. The author concluded that increase in Clostridiales order play a central role in inflammation suppression. But since there are LPS-producing bacteria in Clostridiales order, the increase in short-chain fatty acid-producing bacteria has to be shown.
  5. Overall, the discussion about the mechanism of anti-inflammatory effect was poor.

Author Response

[Comment] The active components in the FPE have to be described in the table. 

[Comment] Relationship between two experiments cannot be simply compared.

[Answer] The authors thank the comment from the reviewer. The content of extract before fermentation, pH, water content, and brix of the extract after fermentation, and lactic acid bacteria used for fermentation were shown in text as according to your comment (Page 2, Line 74-89). Phytochemical function, physiological function, and nutritional compound of FPE produced from 75 kinds of extract reported previously were shown in text (Page4, Line 152-154). Those characteristics of FPE produced from 40 kinds of extract were not analyzed, however, 40 kinds of plant materials were included in FPE produced from 75 kinds of extract, same kinds of lactic acid bacteria were used for fermentation, and total content of fruits, vegetables, and wild herbs, which are sources of prebiotic, was 44 % in extract of 40 kinds and dietary fiber was 4.6 g /100 g extract in FPE from extracts of 40 kinds, which were similar to FPE produced from extract of 75 kinds. These results were shown in text (Page 5, Line 183-189). We showed nutritional compounds in FPE from extract of 40 kinds in Table S3. Our result that change of gut microbiota composition after 3 month-administration of FPE produced from extracts of 40 kinds suggests the physiological effect as same as FPE produced from extracts of 75 kinds and are useful to select the key plant materials for anti-inflammatory effect.

[Comment] The concentrations of fecal short chain fatty acids have to be shown.

[Comment] The increase in short-chain fatty acid-producing bacteria has to be shown.

[Answer] The concentrations of fecal short chain fatty acids was not analyzed. However, increase of Clostridials component in feces after 3 month-administration of FPE suggests the increase of butyric acid because production of butyric acid has been shown in commensal Clostridials, which releases butyric acid as an end-product of dietary fiber fermentation. Furthermore, Clostridiales order in feces after 3-month administration of FPE consisted of 49 % of Lachnospiraceae family and 33 % of Ruminococcaceae family, which produce butyric acid. These results also support the increase of butyric acid. We showed this result in text (Page 6, Line 214-216).

[Comment] The discussion about mechanism of anti-inflammatory effect was poor.

[Answer] According to the comment, the mechanism of anti-inflammatory effect was shown in text (Page 6, Line 216-222).

Round 2

Reviewer 1 Report

In my opinion, Authors tried to correct the manuscript in line with the reviewers' suggestions, and they did it successfully. I have no further comments on the manuscript.

Reviewer 2 Report

Almost all answers were accomplished to several comments from the referee carefully by the author. Preparation of FPE in Materials and Methods was revised properly. I could understand the possible active components in FPE and the speculated mechanism of anti-inflammatory effect. I have no comments for improvements.